# Multilevel analysis of continuation of maternal healthcare services utilization and its associated factors in Ethiopia: A cross-sectional study

**Eshetu E. Chaka** [ORCID] *

Department of Public Health, College of Medicine and Health Sciences, Ambo University, Ambo, Ethiopia

* eejeta@yahoo.com, eshetu@ambou.edu.et

## Abstract

Continuum of care (CoC) has been recognized as a crucial strategy for minimizing maternal, neonatal, and child mortality. CoC promotes integrated Maternal Neonatal and Child Health (MNCH) services by linking together three aspects of maternal health care antenatal care, skilled birth attendance, and postnatal care. The study aimed to assess continuation of maternal healthcare services utilization and its associated factors among reproductive age women at pregnancy, delivery and postnatal stages in Ethiopia. Cross-sectional study design conducted using Ethiopian 2016 Demographic and Health Survey data. All women with the most recent live birth in the last five years preceding the 2016 survey were the study population. The sample size was 7590, 2415, and 1342 at service entry (ANC use), COC at a delivery level, and CoC at Postpartum level respectively. COC was measured at three levels of maternal health care (during pregnancy, delivery, and postpartum). The CoC is constructed from four or more antenatal care visits (ANC4+), skilled birth attendance (SBA), and postnatal care (PNC). About 9.1% of women received all components of CoC. Educational attainment, wealth quintile, and media exposure were associated with four or more antenatal care visits and COC at the delivery level. Perception of getting money for healthcare, having blood pressure measured and urine sample taken during ANC was associated with continuity of care at the delivery level and continuity of care at a postpartum level. Birth order, residence, and region were common factors associated with each outcome of interest. The proportion of women who received all ANC4+, SBA, and PNC across the CoC was low in Ethiopia. Effort needed to increase CoC at each stage. The study shows that focusing on place of residence and regional state variation is necessary to improve CoC at each level. Thus, contextualizing the strategies and further research are critical.

## Introduction

Although the maternal, neonatal and child health problem have attained the top agenda of the international community, they continue to be a serious issue [1, 2]. A worldwide estimate showed that over 5.6 million women and babies died in 2015 due to complications during

**Data Availability Statement:** All relevant data are within the paper and its Supporting Information files.

**Funding:** The authors received no specific funding for this work.

**Competing interests:** The authors have declared that no competing interests exist.

pregnancy, birth and in the first month of life [3]. Almost all maternal deaths occur in a low resource setting of which 62% account for sub-Saharan Africa.

Most deaths occur during labour, delivery and the immediate after delivery and the sad part is that these all deaths could be prevented with proven cost-effective interventions such as antenatal care (ANC), skilled birth attendants (SBA), and postnatal care (PNC). They are considered as crucial maternal health-care services for improving many health outcomes of mothers and babies [4, 5] because these services make sure early detection and management of complications [6]. Given the importance of each of the three maternal health services, recently there was a call for a continuum of care (CoC) for maternal and newborn health emphasizing on continuity of care over time for every woman and integrated service delivery in health facilities [7, 8].

A potent CoC links crucial maternal neonatal and child health (MNCH) interventions across the pregnancy, delivery, and postpartum stages. The advantages of a CoCat each stage builds on the success of the previous stage [9]. Therefore, the CoC is expected to reduce half a million maternal deaths, 4 million neonatal deaths, and 6 million child deaths [7]. A lack of appropriate care at any stage of a CoC is associated with poor MNCH outcomes [10, 11].

Although individual components of MNCH (ANC, SBA, or PNC) services have been studied, none of them have assessed MNCH services along the CoCusing multilevel analysis in Ethiopia. Thus, this study aimed to assess continuation of maternal healthcare services utilization and its associated factors among reproductive age women in Ethiopia.

## Materials and methods

### Ethics statement

This study was approvedby the ethical committee of Tehran University of MedicalSciences (code number: IR.TUMS.SPH.REC.1396.4802).

### Data sources, study populations and sampling

This cross-sectional study analyzed the 2016 Ethiopia Demographic Health Survey (DHS) existing data. The survey used a two-stagesstratified cluster sampling design to select eligible participants.The sampling frame for the survey was the lists of Enumeration areas (EAs) developed from the 2007 population and housing census. In the first stage, 645 (202 from Urban and 443 from rural areas) clusters/EAs were selected from the lists of EAs after stratification of the 11 regional states into urban and rural areas. In the second stage, on average of 24–32 households per cluster/EAs were selected. The survey includes a weighted probability sample of 16,650 households in which 7590 live births reported.Women aged 15–49 years who gave live birthsin the last five years preceding the 2016 surveywere the study population For women who had more than one live birth, only the most recent live birth was considered in this study to get the most recent information and minimize recall bias.

### Variables

**Outcome variables.**   The outcome variable is a continuum of care (CoC). The CoC constructed from four or more antenatal care visits (ANC4+), skilled birth attendance (SBA), and postnatal care (PNC). The SBA was defined as delivery assisted by a health professional (i.e., doctor, nurse/midwife or health officer) in line with the WHO policy guideline [12]. While PNC was defined as the mother's postnatal care at least one visit after delivery at a health facility or at home. Therefore, the three components of CoC are:

1. ANC4+: it was coded as 1 if a woman received ANC4+ for their most recent birth and 0 otherwise. (n = 7590)

2. CoC at the delivery level:it was coded as '1'if women received ANC4+ and SBA and '0' if the women received ANC4+, but did not receive SBA.(n = 2415)

3. CoC at the postpartum level:it was coded as '1' if women received all ANC4+, SBA and PNC and '0' if women received ANC4+ and SBA, but did not receive PNC. (n = 1342)

**Explanatory variables.** Factors affecting each outcome variable were classified as individual and community variables.

Individual variables subgroup as First, Demographic and socio-economic characteristics of women include mother's age at last birth, mother's education, mother's employment, marital status, religious, household wealth index, sex of household head, birth order and pregnancy wanted. Second, Access to healthcare services includes the perception of getting money for healthcare, and getting permission to go to health facility. Lastly, healthcare services utilization include blood pressure has measured, blood sample has taken, urine sample has taken during antenatal care visits, mother received tetanus injection during pregnancy, informed about pregnancy complication and place of delivery. While community variables were included residence, regional state, and distance to health facility

## Statistical analysis

A two-level multilevel binary logistic analysis was employed in order to account for the hierarchical structure of the DHS data and the clustering of response at different levels. It also enables partitioning of the total variation in the outcome into within-group and between-group variances, which allows distinguishing the relative contributions of individual-level and community-level variables [13, 14].The following equation was used to explain the two-level multilevel model for a CoC in which individual woman nested within the community:

$$\text{Logit } (P_{ij}) = \beta 0 + \beta_1 I + \beta_2 C + u_j \tag{1}$$

Where, $P_{ij}$ is the probability of the outcome of interest for woman i in the community j; the β's are the fixed coefficients; I and C refer to individual and community-level independent variables respectively, and $u_j$ indicates the random effects for the $j^{th}$ community.

## Model specification

Model-1was fitted without explanatory variables. Model-2 fitted with individual-level variables. Lastly,model-3, was fitted using both individual-level variables and community-level variables to adjust for both level explanatory variables. All models were fitted sequentially for each outcome variable. Due to spatial considerations, this study reported only model-3.

Fixed effects refer to the individual and community covariates and were reported in terms of odds ratio (OR) with their p-value and 95% confidence intervals. The random effects are measure of variation in outcome variables across community and were reported using community-level variance ($\sigma^2 u$). The intra-cluster correlation Coefficient (ICC) and proportional change in variance (PCV) were used to examine clustering and the extent to which community factors explain the unexplained variance of the empty model. The ICC was calculated as:

$$\text{ICC} = \frac{\sigma u^2}{\sigma u^2 + 3.29} \tag{2}$$

Where:$\sigma^2 u$ is community-level variance (between-community variance) and 3.29 is individual level variance equal to $\pi^2/3$ [15]. A backward variable selection strategy was used to select

independent variables for the final model. A significance level of alpha < 0.05 was used to determine statistical significance. All analysis was performed using STATA 13.

## Ethics consideration

This study is a secondary data analysis of the Ethiopian DHS which is publicly available and permission has received from MEASURE DHS Data Archive at ICF International to conduct this study. After data access is authorized the researcher of this study has maintained the confidentiality of the data. The respondents consent was waived due to the secondary data nature however the survey reported that verbal consent was obtained for their participation. This study was approved by the ethical committee of Tehran University of Medical Sciences.

## Results

### The continuum of care

Nearly 32% of women received ANC4+ visits, 18% CoC at the delivery level and only 9% of women had CoC at a postpartum level among women gave live birth in the last five years preceding the survey in Ethiopia.

### Bi-variate analysis of continuum of care

Table 1 showed that women in the younger age group were more likely to have received ANC4 + than older women. Women resided in the urban area, with higher educational level, and from richest household wealth quintile have received a higher proportion of ANC4+ visits. The highest proportion of ANC4+ visits was found among women living in Addis Ababa or Dire Dawa administrative.

Table 2 showed that CoC at the delivery level increased by women's education and household wealth quintile. Women with birth order 2–3, 4–5 and 6+ were less likely to have CoC at a delivery level compared to those with first birth order. CoC at a delivery level was higher among women resided in urban areas than those residing in rural areas. The lowest proportion

**Table 1. Distribution of ANC4+ visits byselected independent variables among women who had a live birth in the last five years preceding the survey (N = 7590).**

| Variables | ANC4+ | | P-value |
|---|---|---|---|
| | Yes (%) | No (%) | |
| **Mother's age at last birth** | | | 0.001 |
| 15–24 | 30.9 | 69.1 | |
| 25–34 | 34.7 | 65.3 | |
| 35–49 | 27.0 | 73.0 | |
| **Mother's Educational status** | | | 0.001 |
| No education | 24.1 | 75.9 | |
| Primary | 38.5 | 61.5 | |
| Secondary or Above | 66.3 | 33.7 | |
| **Marital status** | | | 0.196 |
| Not married/in union | 35.9 | 64.1 | |
| Married /in union | 31.5 | 68.5 | |
| **Women's employment status** | | | 0.001 |
| Unemployed | 29.6 | 70.4 | |
| Agricultural employed | 31.5 | 68.5 | |

*(Continued)*

**Table 1.** (Continued)

| Variables | ANC4+ | | P-value |
|---|---|---|---|
| | Yes (%) | No (%) | |
| Unskilled manual | 51.8 | 48.2 | |
| Formal employed | 41.6 | 58.4 | |
| **Religion Affiliation** | | | 0.001 |
| Orthodox | 39.0 | 61.0 | |
| Protestant or other | 30.0 | 70.0 | |
| Muslim | 25.7 | 74.3 | |
| **Household wealth quintile** | | | 0.001 |
| Poorest | 18.4 | 81.6 | |
| Poor | 25.3 | 74.7 | |
| Middle | 28.1 | 71.9 | |
| Rich | 36.2 | 63.8 | |
| Richest | 57.4 | 42.6 | |
| **Regional state** | | | 0.001 |
| Oromia | 22.1 | 77.9 | |
| SNNP | 38.2 | 61.8 | |
| Amhara | 31.5 | 68.5 | |
| AA/Dire Dawa | 85.8 | 14.2 | |
| Tigray | 56.5 | 43.5 | |
| Others | 20.8 | 79.2 | |
| **Residence** | | | 0.001 |
| Urban | 62.7 | 37.3 | |
| Rural | 27.3 | 72.7 | |
| **Exposed to media** | | | 0.001 |
| No | 25.0 | 75.0 | |
| Yes | 44.8 | 55.2 | |
| **Sex of Household head** | | | 0.067 |
| Male | 31.2 | 68.8 | |
| Female | 35.5 | 64.5 | |
| **Birth Order of last birth** | | | 0.001 |
| First | 42.0 | 58.0 | |
| 2–3 | 35.4 | 64.6 | |
| 4–5 | 29.9 | 70.1 | |
| 6+ | 22.6 | 77.4 | |
| **Pregnancy Desired** | | | 0.004 |
| Yes | 33.2 | 66.8 | |
| Yes but later | 29.8 | 70.2 | |
| No more | 24.2 | 75.8 | |
| **Getting permission to go to a health facility** | | | 0.001 |
| Big problem | 23.4 | 76.6 | |
| Not a big problem | 36.6 | 63.4 | |
| **Getting money** | | | 0.001 |
| Big Problem | 26.8 | 73.2 | |
| Not a big problem | 39.4 | 60.6 | |
| **Distance to health facility** | | | 0.001 |
| Big problem | 25.1 | 74.9 | |
| Not a big problem | 41.0 | 59.0 | |

**Table 2. Distribution of CoC at the delivery level by selected explanatory variables (N = 2415).**

| Variables | CoC at delivery level | | P-value |
|---|---|---|---|
| | Yes | No | |
| **Mother's age at last birth** | | | 0.078 |
| 15–24 | 61.4 | 38.6 | |
| 25–34 | 54.3 | 45.7 | |
| 35–49 | 52.6 | 47.4 | |
| **Mother's Educational status** | | | 0.001 |
| No education | 40.9 | 59.1 | |
| Primary | 58.2 | 41.8 | |
| Secondary or Above | 89.9 | 10.1 | |
| **Marital status** | | | 0.904 |
| Not married/in union | 54.9 | 45.1 | |
| Married /in union | 55.6 | 44.4 | |
| **Women's employment status** | | | 0.001 |
| Unemployed | 51.7 | 48.3 | |
| Agricultural employed | 52.7 | 47.3 | |
| Unskilled manual | 72.6 | 27.4 | |
| Formal employed | 75.7 | 24.3 | |
| **Religion Affiliation** | | | 0.001 |
| Orthodox | 66.4 | 33.6 | |
| Protestant or other | 42.5 | 57.5 | |
| Muslim | 49.1 | 50.9 | |
| **Household wealth index** | | | 0.001 |
| Poorest | 31.0 | 69.0 | |
| Poor | 39.2 | 60.8 | |
| Middle | 42.5 | 57.5 | |
| Rich | 49.2 | 50.8 | |
| Richest | 87.9 | 12.1 | |
| **Regional state** | | | 0.001 |
| Oromia | 44.3 | 55.7 | |
| SNNP | 46.4 | 53.6 | |
| Amhara | 52.2 | 47.8 | |
| Addis Ababa/Dire Dawa | 94.1 | 5.9 | |
| Tigray | 80.3 | 19.7 | |
| Others | 55.6 | 44.4 | |
| **Residence** | | | 0.001 |
| Urban | 92.5 | 7.5 | |
| Rural | 43.2 | 56.8 | |
| **Exposed to media** | | | 0.001 |
| No | 40.0 | 60.0 | |
| Yes | 72.1 | 27.9 | |
| **Sex of Household Head** | | | 0.006 |
| Male | 53.8 | 46.2 | |
| Female | 64.4 | 35.6 | |
| **Birth order of last birth** | | | 0.001 |
| First | 73.0 | 27.0 | |
| 2–3 | 58.9 | 41.1 | |
| 4–5 | 48.0 | 52.0 | |

(*Continued*)

**Table 2.** (Continued)

| Variables | CoC at delivery level | | P-value |
|---|---|---|---|
| | **Yes** | **No** | |
| 6+ | 36.4 | 63.6 | |
| **Pregnancy desired** | | | 0.995 |
| Yes | 55.6 | 44.4 | |
| Yes but later | 55.5 | 44.5 | |
| No more | 55.1 | 44.9 | |
| **Getting permission to go to a health facility** | | | 0.001 |
| Big problem | 42.0 | 58.0 | |
| Not a big problem | 60.5 | 39.5 | |
| **Getting money** | | | 0.001 |
| Big Problem | 44.9 | 55.1 | |
| Not a big problem | 66.4 | 33.6 | |
| **Distance to health facility** | | | 0.001 |
| Big problem | 41.6 | 58.4 | |
| Not a big problem | 67.5 | 32.5 | |
| **Told about pregnancy complication** | | | 0.001 |
| No | 46.9 | 53.1 | |
| Yes | 63.0 | 37.0 | |
| **Blood pressure measured during ANC visits** | | | 0.001 |
| No | 32.0 | 38.0 | |
| Yes | 61.2 | 38.8 | |
| **A blood sample taken during ANC visits** | | | 0.001 |
| No | 27.8 | 72.2 | |
| Yes | 62.4 | 37.6 | |
| **A urine sample taken during ANC visits** | | | 0.001 |
| No | 28.2 | 71.8 | |
| Yes | 64.2 | 35.8 | |
| **Received tetanus injection during ANC** | | | 0.277 |
| No | 52.2 | 47.8 | |
| Yes | 56.5 | 43.5 | |

of CoC at the delivery level was found in Oromia regional state and the highest was in Addis Ababa.

Table 3 showed that formally employed women were more likely to CoC at a postpartum level than those women agriculturally employed. A lowerproportionofCoC at postpartum level was found in Oromia regional state compared to Tigray regional state and Addis Ababa.

Marital status and sex of household head were significantly associated with CoC at the postpartum level, with married women and male-head household exhibiting the lower CoC at a postpartum level compared to their counterparts. The likelihood of CoC at postpartum level was higher among women gave birth in the health facility compared to those gave birth at home.

The antenatal care quality indicators were significantly associated with the completion of CoC. For example, the CoC at postpartum level is higher among women that had their blood pressure measured during antenatal care than not measured.

**Table 3. Distribution of Completion CoC at the postpartum level by explanatory variables (N = 1342).**

| Variables | COC at postpartum level | | |
|---|---|---|---|
| | **Yes** | **No** | **P-value** |
| **Mother's age at last birth** | | | 0.369 |
| 15–24 | 47.4 | 52.6 | |
| 25–34 | 54.0 | 46.0 | |
| 35–49 | 50.6 | 49.4 | |
| **Mother's Educational status** | | | 0.672 |
| No education | 49.8 | 50.2 | |
| Primary | 51.5 | 48.5 | |
| Secondary or Above | 54.0 | 46.0 | |
| **Marital status** | | | 0.038 |
| Not married/in union | 65.5 | 34.5 | |
| Married | 50.6 | 49.4 | |
| Women's employment status | | | 0.057 |
| Unemployed | 51.8 | 48.2 | |
| Agricultural employed | 46.0 | 54.0 | |
| Unskilled manual | 53.0 | 47.0 | |
| Formal employed | 62.4 | 37.6 | |
| **Religion Affiliation** | | | 0.012 |
| Orthodox | 57.4 | 42.6 | |
| Protestant or other | 46.5 | 53.5 | |
| Muslim | 43.0 | 57.0 | |
| **Household wealth index** | | | 0.091 |
| Poorest | 48.2 | 51.8 | |
| Poor | 45.6 | 54.4 | |
| Middle | 56.7 | 43.3 | |
| Rich | 42.5 | 57.5 | |
| Richest | 55.9 | 44.1 | |
| **Regional state** | | | 0.001 |
| Oromia | 34.5 | 65.5 | |
| SNNP | 48.7 | 51.3 | |
| Amhara | 48.7 | 51.3 | |
| Addis Ababa& Dire Dawa | 64.4 | 35.6 | |
| Tigray | 64.5 | 35.5 | |
| Others | 54.4 | 45.6 | |
| **Residence** | | | 0.174 |
| Urban | 54.8 | 45.2 | |
| Rural | 49.3 | 50.7 | |
| **Exposed to media** | | | 0.088 |
| No | 46.8 | 53.2 | |
| Yes | 54.5 | 45.5 | |
| **Sex of Household Head** | | | 0.018 |
| Male | 49.4 | 50.6 | |
| Female | 61.1 | 38.9 | |
| **Birth order of last birth** | | | 0.432 |
| First | 47.7 | 52.3 | |
| 2–3 | 55.5 | 44.5 | |
| 4–5 | 50.9 | 49.1 | |

(*Continued*)

**Table 3.** (Continued)

| Variables | COC at postpartum level | | |
|---|---|---|---|
| | Yes | No | P-value |
| 6+ | 52.0 | 48.0 | |
| **Place of delivery** | | | 0.009 |
| Home | 23.3 | 76.7 | |
| Health facility | 52.4 | 47.6 | |
| **Pregnancy Desired** | | | 0.247 |
| Yes | 50.7 | 49.3 | |
| Yes but later | 58.1 | 41.9 | |
| No more | 46.5 | 53.5 | |
| **Getting permission to go to health facility** | | | 0.001 |
| Big problem | 38.1 | 61.9 | |
| Not a big problem | 55.0 | 45.0 | |
| **Getting money** | | | 0.007 |
| Big Problem | 44.9 | 55.1 | |
| Not a big problem | 56.3 | 43.7 | |
| **Distance to health facility** | | | 0.039 |
| Big problem | 45.8 | 54.2 | |
| Not a big problem | 54.6 | 45.4 | |
| **Told about pregnancy complication** | | | 0.001 |
| No | 40.9 | 59.1 | |
| Yes | 58.4 | 41.6 | |
| **Blood pressure measured during ANC** | | | 0.001 |
| No | 20.0 | 80.0 | |
| Yes | 55.6 | 44.4 | |
| **A blood sample is taken during ANC** | | | 0.001 |
| No | 32.0 | 68.0 | |
| Yes | 53.7 | 46.3 | |
| **A urine sample is taken during ANC** | | | 0.001 |
| No | 33.2 | 66.8 | |
| Yes | 54.2 | 45.8 | |
| **Received tetanus injection during ANC** | | | 0.003 |
| No | 40.8 | 59.2 | |
| Yes | 54.3 | 45.7 | |

## Multi-level model analysis

The result of the random effect in each model was shown in Table 4. There were significant community-level variations in each conditional component of CoC across communities. For example, in model-1 (intercept-only model) ICC showed considerable variation in the CoC across communities (24.0% - 58.3%), which is due to cultural, social and economic differences across communities. The ICC in the adjusted multilevel model-3 reduced to 16.3%, 33.8%, and 20.4% in the ANC4+, CoC at the delivery level and CoC at postpartum level respectively. However ICC decreased along each model, the variation remained significant. This indicates the presence of unobserved factors in the study.

At the individual-level variables, odds of ANC4+ visitswas 1.51 times and 1.33 times higher among women aged 25–34 and 35–49 compared to those aged 15–24 respectively. Similarly, the odds of ANC4+ visit were increased by the mother's education level and wealth quintile. However, odds of ANC4+ visits decreased along with birth order (Table 5).

**Table 4. Random effect estimates of multilevel model for each outcome variable.**

| Random effect | Model-1 | Model-2 | Model-3 |
|---|---|---|---|
| **ANC4+** | | | |
| Community random Variance (SE) | 1.67(0.18)*** | 0.89(0.10)*** | 0.64(0.08)*** |
| ICC | 33.7% | 21.3% | 16.3% |
| PCV | Reference | 46.8% | 61.7% |
| **CoC at delivery level** | | | |
| Community random Variance (SE) | 4.60(0.64)*** | 1.89(0.31)*** | 1.68(0.29)*** |
| ICC | 58.3% | 36.5% | 33.8% |
| PCV | Reference | 58.9% | 63.5% |
| **CoC at postpartum level** | | | |
| Community random Variance (SE) | 1.03(0 .22)*** | 0.95(0.23)*** | 0.84(0.21)*** |
| ICC | 24.0% | 22.4% | 20.4% |
| PCV | Reference | 7.8 | 18.4 |

SE = Standard error ICC = Intra-class correlation PCV = Proportion of change in variance

*** p<0.001

At the individual-level variables, mother attended secondary or higher education was 2.71 times more likely to have CoC at delivery level (95%CI: 1.62–4.53) as compared with no education. Birth order was significantly associated with CoC at the delivery level. The finding also shows that married women were more likely to CoC at the delivery level than not married/in the union.

Antenatal care quality indicators were significantly associated with CoC at the delivery level. The odds of CoC at delivery level was 58.0% higher if blood pressure was measured (OR = 1.58; 95%CI: 1.12–2.23) and twice more likely if urine sample was taken (OR = 2.04; 95%CI: 1.48, 2.80) during antenatal care compared with their counterparts (Table 5).

The household wealth quintile and exposure to media showed a significant positive association with CoC at the delivery level. Women from richest wealth quintile were 2.60 times significantly more likely (95%CI: 1.44, 4.70) to CoC at a delivery level compared with women from poorest wealth quintile. The odds of CoC at the delivery level was 1.42 times more likely among women exposed to media than not exposed (Table 5).

Maternal age at last birth, religious affiliation, sex of household head, distance to health facility, told about pregnancy complication and a blood sample was taken were not significantly associated with CoC at the delivery level.

At the Community-level, the odds of CoC at the delivery level was 84% less likely among women from rural areas while it was 7 times more likely among women from Tigray regional state compared to their counterparts (Table 5).

At the individual level, mother's employment, place of delivery and birth order were significantly associated with the CoC at postpartum level. The odds of CoC at postpartum level was 1.56 times higher among unemployed women and 2.04 times among unskilled employed women compared to those agriculturallyemployed.Similarly, CoCat postpartum level was 4.85 times more likely among women who gave birth in a health facility than women gave birth at home. Women with 2–3, 4–5 and 6+ birth order were more likely to have CoC at a postpartum level than first birth order (Table 5).

Sex of household head and getting money for healthcare services were also associated with the CoC at postpartum level. Women from female-headed households were 58% more likely to have CoC at a postpartum level as compared to women from male-headed households. The

**Table 5. Multivariate multilevel logistic model of achievement of ANC4+ visits; CoC at delivery level; and CoC at postpartum level and adjusted odds ratio with a 95% confidence interval.**

| Variables | ANC4+ | CoC at deliver level | CoC at Postpartum level |
|---|---|---|---|
| Fixed effect | AOR (95%CI) | AOR (95%CI) | AOR (95%CI) |
| **Individual-level variables** | | | |
| **Mother's age at last birth** | | | |
| 15–24 | 1.00 | 1.00 | 1.00 |
| 25–34 | 1.51(1.26, 1.80)*** | 1.13 (0.86,1.46) | 1.07(0.72, 1.59) |
| 35–49 | 1.33 (1.04, 1.70)* | 1.27 (0.75, 2.14) | 0.73(0.41, 1.30) |
| **Mother's Educational status** | | | |
| No education | 1.00 | 1.00 | |
| Primary | 1.61 (1.39, 1.87)*** | 1.12(0.83, 1.51) | |
| Secondary or Above | 2.57(1.98,3.32)*** | 2.71(1.62, 4.53)** | |
| **Marital status** | | | |
| Not married/in union | | 1.00 | 1.00 |
| Married /in union | | 2.02(1.20, 3.40)** | 0.57(0.32, 1.02) |
| Women's employment status | | | |
| Agricultural employed | | | 1.00 |
| Unemployed | | | 1.56(1.12, 2.17)** |
| Unskilled manual | | | 2.04(1.02, 4.05)* |
| Formal employed | | | 2.14(1.37, 3.35)** |
| **Religion Affiliation** | | | |
| Orthodox | 1.00 | | 1.00 |
| Protestant or other | 0.83(0.65, 1.07) | | 1.43(0.84, 2.45) |
| Muslim | 1.02(0.81,1.29) | | 0.82(0.53, 1.25) |
| **Exposed to media** | | | |
| No | 1.00 | 1.00 | |
| Yes | 1.15(1.00, 1.33)* | 1.42(1.07, 1.89)* | |
| **Household wealth quintile** | | | |
| Poorest | 1.00 | 1.00 | 1.00 |
| Poor | 1.35(1.10, 1.65)** | 1.37(0.89, 2.12) | 1.06(0.55, 2.05) |
| Middle | 1.42(1.15,1.75)** | 1.31(0.84,2.04) | 1.88(0.97, 3.66) |
| Rich | 1.89(1.52. 2.35)*** | 1.15(0.72,1.83) | 0.87(0.46,1.66) |
| Richest | 2.06(1.55, 2.74)*** | 2.60(1.44,4.70)** | 1.70(0.84, 3.42) |
| **Sex of household head** | | | |
| Male | | | 1.00 |
| Female | | | 1.58 (1.08, 2.31)* |
| **Birth order of last birth** | | | |
| First | 1.00 | 1.00 | 1.00 |
| 2–3 | 0.81(0.68, 0.98)* | 0.42(0.29,0.61)*** | 1.78(1.22, 2.54)** |
| 4–5 | 0.89(.71, 1.12) | 0.44(0.28, .70)** | 2.29(1.37, 3.84)** |
| 6+ | 0.70(.54, 0.91)** | 0.32(0.18,0.55)*** | 2.47(1.31, 4.69)** |
| **Pregnancy wanted** | | | |
| Yes | 1.00 | | |
| Yes but later | 0.63(0.53, 0.74)*** | | |
| No more | 0.64(0.51, 0.80)*** | | |
| **Told about pregnancy complication** | NA | | |
| No | | 1.00 | 1.00 |
| Yes | | 1.19(0.92, 1.54) | 1.57(1.16,2.11)** |

*(Continued)*

**Table 5.** (Continued)

| Variables | ANC4+ | CoC at deliver level | CoC at Postpartum level |
|---|---|---|---|
| **Fixed effect** | **AOR (95%CI)** | **AOR (95%CI)** | **AOR (95%CI)** |
| **Blood pressure measured during ANC visits** | NA | | |
| No | | 1.00 | 1.00 |
| Yes | | 1.58(1.12, 2.23)** | 4.31(2.47,7.52)*** |
| **Urine sample taken during ANC visits** | NA | | |
| No | | 1.00 | 1.00 |
| Yes | | 2.04(1.48, 2.80)*** | 1.66(1.02, 2.74)* |
| **Received tetanus injection during ANC** | NA | | |
| No | | | 1.00 |
| Yes | | | 2.04(1.42, 2.92)*** |
| **Place of delivery** | NA | NA | |
| Home | | | 1.00 |
| Health facility | | | 4.85(1.75, 13.37)** |
| **Getting Permission** | | | |
| Big Problem | 1.00 | | |
| Not a big problem | 1.12(0.96, 1.31) | | |
| **Getting money** | | | |
| Big Problem | | 1.00 | 1.00 |
| Not a big problem | | 1.36(1.03, 1.80)* | 1.40(1.03,1.90)* |
| **Community-level variables** | | | |
| **Distance to health facility** | | | |
| Big problem | 1.00 | 1.00 | |
| Not a big problem | 1.08(0.93, 1.25) | 1.18(0.87, 1.60) | |
| **Residence** | | | |
| Urban | 1.00 | 1.00 | 1.00 |
| Rural | 0.50(0.35, 0.72)*** | 0.16(0.08, .33)*** | 2.07(1.14, 3.76)* |
| **Regional state** | | | |
| Oromia | 1.00 | 1.00 | 1.00 |
| SNNP | 2.44(1.75, 3.39)*** | 1.60(0.90,2.84) | 1.67(0.89, 3.15) |
| Amhara | 1.33(0.94,1.87) | 1.07(0.58, 1.98) | 2.43(1.26, 4.67)** |
| Addis Ababa/Dire Dawa | 7.12(4.12,12.30)*** | 3.30 (1.22,8.91)* | 3.63(1.83, 7.20)*** |
| Tigray | 4.51 (3.04, 6.70)*** | 7.09(3.59, 13.99)*** | 3.63(1.88,7.01)*** |
| Others | 0.91(0.63, 1.33) | 1.46(.69,3.08) | 2.54 (1.13,5.72)* |
| **Model fit statistics** | | | |
| Deviance | 7859.21 | 2205.95 | 1579.90 |
| AIC | 7911.21 | 2255.95 | 1639.89 |

*p<0.05

**p< 0.01

*** p<0.001

Note SNNP = Southern Nation and Nationalities People, AIC = Akaike Information Criterion, AOR = Adjusted Odds ratio

likelihood of CoC at postpartum level was 40% more likely among women who can get money for maternal health care compared to those had a difficult to get money.

The antenatal care quality indicators were significantly associated with the CoC at postpartum level. For example, CoC at postpartum level was 4.31 times and 2.04 times more likely among women who had blood pressure measured and if women received tetanus injection

during ANC visits. Further, the odds of CoC at postpartum level was 57% higher if a woman was told about pregnancy complication during antenatal care visits (Table 5).

Individual-level variables such as mother's education, household wealth index, religion, marital status, exposure to media, and maternal age at last birth were not significantly associated with CoC at postpartum level.

Community-levelvariables, place of residence and regional state were significantly associated with the CoC at postpartum level. After controlling for the contribution of all the individual- and community-level factors, surprisingly the odds of CoC at postpartum level was twice higher among women residing in a rural area compared to reside in an urban area.

## Discussion

The purpose of this is to assess the continuation of maternal healthcare services and associated factors associated at pregnancy, delivery and postpartum level.

Nearly 32.0% of women received ANC4+, 18.0% have COC at delivery level, and only 9.1% have CoC at a postpartum level in Ethiopia. The results highlighted that educational attainment, wealth quintile, and media exposure were associated with four or more antenatal care visits and continuity of care at the delivery level. Perception of getting money for healthcare, had blood pressure measured and urine sample taken during antenatal care was associated with continuity of care at the delivery level and continuity of care at a postpartum level. The study identified that birth order, residence, and regional state were the common factors significantly associated with each three outcome variables.

The results showed that ANC4+ was significantly associated with the mother's age and whether pregnancy planned. Previous studies have also reported that the odds of ANC4+visits higher among women with planned pregnancy and older age [16, 17]. In the study, most educated women and those from higher wealth quintile were more likely to have ANC4+ visits. The possible explanation is that educated women usually have high knowledge about ANC and more access to get ANC services. Moreover, education empower women to control their health care and encourage to access quality maternal health care. This finding was consistent with the previous study [16–20]. However, it was inconsistent with other study reported the reverse direction of association [21].

Birth order and residence were significantly associated with ANC4+ visits. This is consistent with previous studies that reportedwomen with 2–3 and 6+ birth order were less likely to have ANC4+ visits as compared to the first birth order [16, 17, 22] because women might consider themselves as they have sufficient knowledge and experience with pregnancy. Moreover, women resided in rural areas were less likely to have ANC4+ visits in line with the previous studies [18, 23, 24]. However, some previous studies failed to report a significant association between residence and ANC4+ visits [16, 17, 22, 25].

In this study, CoC at the delivery level was significantly associated with the mother's education and wealth quintile. This finding is consistent with previous studies that reported increased CoC at the delivery level by educational level and wealth quintile [18, 26, 27]. However, one study reported none significant association [28].

Women with 2–3, 4–5 and 6+ birth order were less likely to have CoC at the delivery level. This finding congruent with the previous studies in which women with first birth order were more likely to have CoC at delivery level [18, 26, 27]. These women might have trouble accessing services because of childcare, having fewer resources due to large family size or their maternal health care experiences from a previous birth. In contrast to this finding, one study reported the absence of association [28].

Many other previous studies reported that there were significant disparities in the CoC at delivery level by regional state and place of residence [18, 26, 27]. Women resided in rural areas and living in a region with a dispersed population were less likely to continued care up to the delivery level as our study reported. Some of the reasons suggested include the rural area's dispersed nature, a limited supply of health facilities, and a long distance from health services.

Exposed to media and perception of getting money for healthcare service were increased the odds of CoC at the delivery level and supported by previous studies elsewhere [18, 27]. Women's knowledge of health care needs during birth, as well as the availability and accessibility of such services, is influenced by media exposure. Furthermore, media exposure is critical for bridging knowledge gaps and dispelling any myths or misconceptions that may exist.

The study also showed that women who had blood pressure measured and urine sample have taken during antenatal care visits have increased odds of CoC at the delivery level. This findings supported bythe previous study and argued that women with high-quality antenatal care become better informed about pregnancy and more likely to recognize the importance of skilled delivery care [26].

In contrast to prior studies, CoC at the postpartum levelwas significantly associated with women's employment status [9, 18].In line with other studies, mother's education status [26, 29, 30], wealth quintile [29], and distance from health facility [26, 29]were not significantly associated with CoC at postpartum level as other studies have found.

However, being from higher wealth quintile[18, 26–28, 31],having a highereducational level [18, 27, 28, 31], and being exposed to mass media [18, 28, 30]were significantly associated with CoCat a postpartum level in other studies. An educated woman able to make better maternla health decisions and more likley to have the financial to pay for maternla health care without having rely on their husbands.

In line with prior study conducted in Cambodia [9, 26, 29], the mother's age at last birth was not significantly associated with the CoC at a postpartum level in this study. The findings, however, contradict those of studies conducted in Pakistan and Ghana [18, 30].Previous studies have shown that women who had birth in a health facility [26, 28] and perceived getting money for healthcare services [27] were more likely to have CoC at the postpartum level in this study.

In terms of birth order, several prior studies found no association between birth order and CoC at postpartum level [26–28], however, a study in Pakistan found that women with lower birth order were more likely to have CoC at postpartum level [18]. In this study, however, women with a higher birth order (> 3 births) were more likely to have CoC at the postpartum level.

Women who received maternalhealthcare service during ANC visits were more likely to have CoC at the postpartum level. Other prior studies [26, 28] backed up this conclusion. In reality, women are better informed about pregnancy and understand the importance of each service provided.

At the communitylevel, women's place of residence and regional state were significantly associated with the COC at the postpartum level. However, the place of residence was not significantly associated with CoC at a postpartum level in the previous study [18, 26, 27]. Regional variation in CoC at the postpartum level reported in previous study [18, 27] due to the dispersed population over a larger area, poor infrastructure and no access to healthcare services.

The study main strength is that it used nationally representative data to inform planners and decision makers about how to strengthen the continuum of maternal health care in Ethiopia. This study, however, has a few limitations. First, the study used a cross-sectional study design and collected data of last five years which is prone to recall bias. Second, it employed primary sampling unit (PSU) as a proxy of community boundaries. The use of the DHS

primary sampling unit as the community boundary may lead to selection bias. Finally, this study did not cover all relevant factors (e.g. quality of healthcare services).

## Conclusion

The proportion of women who received all ANC4+, SBA, and PNC across the CoC was low in Ethiopia. This study indicated the need for an effort to increase the continuum of care by promoting the next stage services during each previous stage. Birth order, place of residence, and regional state were the common factors significantly associated with the components of CoC at each level, whereas other factors were specific to each level of continuum of care. Therefore, the study suggests that a need for strategies that should be targeted at specific factors promoting CoC. This study also revealed the necessity to contextualize the strategies and explore the factors responsible for the unexplained variance in CoC. Thus, more study is needed to identify those factors.

## Acknowledgments

The authors thank all the study participants, interviewers and Measure DHS project for providing the datasets for this analysis.

## Author Contributions

**Conceptualization:** Eshetu E. Chaka.

**Formal analysis:** Eshetu E. Chaka.

**Methodology:** Eshetu E. Chaka.

**Writing – original draft:** Eshetu E. Chaka.

**Writing – review & editing:** Eshetu E. Chaka.

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
