## [Decision Letter · Decision Letter 0]

15 Feb 2022

PGPH-D-21-00920

Multilevel Analysis of Continuation of Maternal Healthcare Services Utilization and Its Associated Factors in Ethiopia: A Cross-Sectional Study

Dear Dr. Chaka,

Thank you for submitting your manuscript to PLOS Global Public Health. After careful consideration, we feel that it has merit but does not fully meet PLOS Global Public Health’s publication criteria as it currently stands. Therefore, we invite you to submit a revised version of the manuscript that addresses the points raised during the review process.

We look forward to receiving your revised manuscript.

Kind regards,

Zohra S. Lassi, PhD

Academic Editor

Journal Requirements:

1. Please update the completed 'Competing Interests' statement, including any COIs declared by your co-authors. If you have no competing interests to declare, please state "The authors have declared that no competing interests exist".

Additional Editor Comments (if provided):

Reviewers' comments:

Reviewer's Responses to Questions

**Comments to the Author**

1. Does this manuscript meet PLOS Global Public Health’s publication criteria? Is the manuscript technically sound, and do the data support the conclusions? The manuscript must describe methodologically and ethically rigorous research with conclusions that are appropriately drawn based on the data presented.

Reviewer #1: Yes

Reviewer #2: Partly

Reviewer #3: Yes

Reviewer #4: Yes

2. Has the statistical analysis been performed appropriately and rigorously?

Reviewer #1: Yes

Reviewer #2: Yes

Reviewer #3: No

Reviewer #4: Yes

3. Have the authors made all data underlying the findings in their manuscript fully available (please refer to the Data Availability Statement at the start of the manuscript PDF file)?

Reviewer #1: Yes

Reviewer #2: Yes

Reviewer #3: Yes

Reviewer #4: Yes

4. Is the manuscript presented in an intelligible fashion and written in standard English?

Reviewer #1: Yes

Reviewer #2: Yes

Reviewer #3: No

Reviewer #4: Yes

5. Review Comments to the Author

Reviewer #1: In the abstract: Check the use of acronyms please check conclusions in the abstracts as it is difficult to understand "targeting residence" ,as it is not clear from the results

In the introduction please clarify that in Ethiopia until the best of the knowledge CoC has not been studied

Material and Methods: Please state if the survey was conducted during 2016 and addressess women with live births five years earlier this means 2011-2016 ? Please define clearly the period. Also, it is not clear wether you used informetion that already existed in the survey or tracked eligible participants and obtained the information after 2016

A more in detail description of the survey and the information collected would be great to better understand what was done and how

Out come and Explanatory variables would benefit of a more graphical display (Table or graph )

Results: What is higher birth order? number of children? please clarify what you are refering !

Discussion: Better start with the study results

Check english for example: line 232 in page 11 ANC+ visits are higher (include the verb please!)

in line 232 and 233 instead of pregnancy wanted should be planned pregnancies right ?

Please describe better the comparison studies for example XXX et al found that in contrast our study found ....

Again describe birth order as previously said

The discussion should be enriched by the comments of the author on what is the meaning of all the correlations described and the fact that variable results are to be found among the different studies, for example: why the place of residence is not associated with CoC at a postpartum level whereas other variables might be this contributes to highlight the importance of CoC overall, maybe is because women that have CoC at post partum level are aware of the key relevance of this regardless of the place of residence (far away or near the health care facility???)

What are the strenghts of the study? Only limitations ?

Reviewer #2: The authors investigated the factors associated with the continuum of care (CoC) in Ethiopia. In general, the concept of the manuscript is of a great interest and the readership of the Journal may benefit from the presented facts and study results. However, this paper needs to improve in order to make the main message of the study clearer to the reader. The research method is poorly described. Some concerns for improvement of the paper are as follows:

Introduction

- The authors need to address why specifically this study is needed, as there remains a plethora of studies, particularly in the factors associated with the individual component of continuation of maternal health service utilization.

-Lines 74-75, the study objective is unclear. What do you mean by the level?

Materials and methods

- Please describe the study population more thoroughly. How many participants were reported in the DHS data, and how many were included in your study? The authors should describe how they reached the final sample size for their analysis. The authors may want to rethink the structure of the method section to help the readers to develop a good understanding of the study population.

- Please provide more details about stratified two-stage cluster sampling design of Ethiopian DHS 2016 and provide a reference for the study design. Provide more details about each stage, such as what are the sampling units, how many units are selected and out of how many, how are those units selected.

- Outcome variables: What is the basis of defining three components of CoC? Provide the references or the basis for the definition.

- Statistical analysis: It is not clear what you are referring to two-level multilevel logistic analysis. What are the different levels? The DHS has a complex survey design; however, your data analysis seems to ignore the survey design. Please provide more details about the statistical analysis.

Discussion

- In general, this section does not clearly present the authors’ conclusions and make some broad generalizations that are not based on the data presented. The author should discuss their findings, mainly why the factors associated with CoC are particularly important in Ethiopia.

- The authors should elaborate the study limitations.

Reviewer #3: 1. A thorough English Editing is required.

2. From Table 1 to 3 Sex of Household should b as Household dead

3. All tables Women Employee status should be correct as Women's Employment Status; most tables need to define p values in notes.

4. Authors have not used various interactions of independent variables e.g. mother education interact with wealth index.

5. Authors need to construct a conceptual framework and then provide enough reasoning for inclusion/exclusion of independent variables in the model.

6. I think multinomial logistic regression model is more appropriate to see ANC and then TBA effect.

7. It is not clear Whether skilled birth attendance (SBA) include Institutional Delivery i.e. took place at a medical Centre/hospital. If a very large proportion of births are institutional then SBA or TBA term is very misleading.

8. Discussion needs a great deal of work.

9. Also they need to elaborate what is the difference between continuation and continuum care.

Reviewer #4: Can be published without modifications. The paper investigates the level of continuity of maternal care and identify the

30 factors affecting the CoC in Ethiopia. It is a well written and well analyzed paper.

6. PLOS authors have the option to publish the peer review history of their article (what does this mean?). If published, this will include your full peer review and any attached files.

**Do you want your identity to be public for this peer review?** For information about this choice, including consent withdrawal, please see our Privacy Policy.

Reviewer #1: **Yes: **Marianella Herrera-Cuenca

Reviewer #2: **Yes: **Dirga Kumar Lamichhane

Reviewer #3: **Yes: **Anil Gumber

Reviewer #4: No

---

## [Decision Letter · Decision Letter 1]

3 May 2022

Multilevel Analysis of Continuation of Maternal Healthcare Services Utilization and Its Associated Factors in Ethiopia: A Cross-Sectional Study

PGPH-D-21-00920R1

Dear Dr Chaka,

We are pleased to inform you that your manuscript 'Multilevel Analysis of Continuation of Maternal Healthcare Services Utilization and Its Associated Factors in Ethiopia: A Cross-Sectional Study' has been provisionally accepted for publication in PLOS Global Public Health.

Best regards,

Zohra S. Lassi, PhD

Academic Editor

Reviewer Comments (if any, and for reference):

Reviewer's Responses to Questions

**Comments to the Author**

1. If the authors have adequately addressed your comments raised in a previous round of review and you feel that this manuscript is now acceptable for publication, you may indicate that here to bypass the “Comments to the Author” section, enter your conflict of interest statement in the “Confidential to Editor” section, and submit your "Accept" recommendation.

Reviewer #1: All comments have been addressed

Reviewer #2: All comments have been addressed

Reviewer #4: All comments have been addressed

2. Does this manuscript meet PLOS Global Public Health’s publication criteria? Is the manuscript technically sound, and do the data support the conclusions? The manuscript must describe methodologically and ethically rigorous research with conclusions that are appropriately drawn based on the data presented.

Reviewer #1: Yes

Reviewer #2: Yes

Reviewer #4: Yes

3. Has the statistical analysis been performed appropriately and rigorously?

Reviewer #1: Yes

Reviewer #2: Yes

Reviewer #4: Yes

4. Have the authors made all data underlying the findings in their manuscript fully available (please refer to the Data Availability Statement at the start of the manuscript PDF file)?

Reviewer #1: Yes

Reviewer #2: Yes

Reviewer #4: (No Response)

5. Is the manuscript presented in an intelligible fashion and written in standard English?

Reviewer #1: No

Reviewer #2: Yes

Reviewer #4: Yes

6. Review Comments to the Author

Reviewer #1: Please check in the discussion lines 306 to 309 because this is not clear and has many repeated words

Reviewer #2: (No Response)

Reviewer #4: (No Response)

7. PLOS authors have the option to publish the peer review history of their article (what does this mean?). If published, this will include your full peer review and any attached files.

**Do you want your identity to be public for this peer review?** For information about this choice, including consent withdrawal, please see our Privacy Policy.

Reviewer #1: **Yes: **Marianella Herrera-Cuenca

Reviewer #2: **Yes: **Dirga Kumar Lamichhane

Reviewer #4: No
